Atypical functioning of female genitalia explains monandry in a butterfly

Xochipiltecatl David 1
Baixeras Joaquín 2
Cordero Carlos R. cordero@ecologia.unam.mx crafaelcm@gmail.com 3
1 Posgrado en Ciencias Biológicas, Instituto de Ecología, Universidad Nacional Autónoma de México , Mexico City , México
2 Institut Cavanilles de Biodiversitat i Biologia Evolutiva, Universitat de València , Paterna , Valencia , Spain
3 Departamento de Ecología Evolutiva, Universidad Nacional Autónoma de México , Mexico City , México
Gillespie Joseph
Electronic publication date: 2021 Nov 22
Publication date: 2021
Volume: 9
Electronic Location ID: e12499
Received 2021 Jul 13; Accepted 2021 Oct 26
Copyright: ©2021 Xochipiltecatl et al.
Copyright year: 2021
Copyright holder: Xochipiltecatl et al.
License: This is an open access article distributed under the terms of the Creative Commons Attribution License, which permits unrestricted use, distribution, reproduction and adaptation in any medium and for any purpose provided that it is properly attributed. For attribution, the original author(s), title, publication source (PeerJ) and either DOI or URL of the article must be cited.
License URL: https://creativecommons.org/licenses/by/4.0/

Keywords: Female genitalia, Spermatophore, Mating frequency, Monandry, Sexual selection, Lepidoptera

Funding: PAPIIT-DGAPA, UNAM IN214120 PAEP, UNAM This research was supported by a grant from PAPIIT-DGAPA, UNAM (IN214120) to Carlos Cordero. David Xochipiltecatl García had a CONACYT scholarship during his graduate studies and his research in the University of Valencia, Spain, was supported by PAEP, UNAM. The funders had no role in study design, data collection and analysis, decision to publish, or preparation of the manuscript.

==============================
Monandrous species are rare in nature, especially in animals where males transfer nutrients to females in the ejaculate. The proximate mechanisms responsible for monandry are poorly studied. In butterflies and moths, the male transfers a nutritious spermatophore into the corpus bursae (CB) of the female. The CB is a multifunctional organ that digests the spermatophore and has partial control of the post-mating sexual receptivity of the female. The spermatophore distends the CB and the post-mating sexual receptivity of the female is inversely proportional to the degree of distension. The CB of many butterfly species has a muscular sheath whose contractions mechanically contribute to digest the spermatophore. As the contents of the CB are absorbed, the degree of distension decreases and the female recovers receptivity. We studied the monandrous butterfly Leptophobia aripa (Boisduval, 1836) (Pieridae) and found that females do not digest the spermatophores. We investigated the structure of the CB and found that a muscular sheath is absent, indicating that in this butterfly females lack the necessary “apparatus” for the mechanical digestion of the spermatophore. We propose that female monandry in this species is result of its incapability to mechanically digest the spermatophore, which results in a constant degree of CB distension after mating and, thus, in the maintenance of the sexually unreceptive state of females. Hypotheses on the evolution of this mechanism are discussed.

Introduction

Monandrous species, in which most females copulate just with one male, are rare in most animal groups (Pizzari & Wedell, 2013; Taylor, Price & Wedell, 2014). There are two general hypotheses to explain the existence of monandry. First, monandry could be selected for when females maximize their fitness with just one mating, which could happen if, for example, polyandry imposes high costs on females (Arnqvist & Andrés, 2006; Jiggins, 2017). Second, if polyandry increases female fitness, sperm competition could favour male adaptations that impose monandry and, in consequence, fitness costs on females (Hosken et al., 2009). Different female adaptations are expected to evolve in each case. For example, if monandry is adaptive for females, they could evolve structures that facilitate the deposition and storage of male-derived anti-aphrodisiacs, as in Heliconius butterflies (Jiggins, 2017). On the other hand, if males impose monandry selection could favour the evolution of counter-adaptations, female traits that prevent or reduce male manipulation, as the anti-antiaphrodisiacs of the plant bug Lygus hesperus (Brent, Byers & Levi-Zada, 2017). These examples show that understanding the proximate mechanisms preventing female remating sheds light on the ultimate causes of monandry.

During copulation, male lepidopterans transfer ejaculates, mostly contained within a spermatophore, into a bag-shaped organ of the female reproductive tract known as corpus bursae (CB hereafter) (Drummond III, 1984; Watanabe & Sato, 1993; Watanabe, 2016; Meslin et al., 2017). In most butterflies and moths studied to date, the ejaculates are rich in nutrients (Boggs & Gilbert, 1979; Marshall, 1985; Pivnick & McNeil, 1987; Boggs, 1990; Lai-Fook, 1991; Watanabe & Sato, 1993; Bissoondath & Wiklund, 1995; Bissoondath & Wiklund, 1996a; Bissoondath & Wiklund, 1996b; Karlsson, 1998; Molleman et al., 2005; Watanabe, 2016; Meslin et al., 2017; Cannon, 2020) and other chemical compounds (Dussourd et al., 1988; Dussourd et al., 1989; Eisner & Meinwald, 1995; Smedley & Eisner, 1996; Cardoso, Roper & Gilbert, 2009; Watanabe, 2016) that enhance female fitness (Vahed, 1998; Arnqvist & Nilsson, 2000; Oberhauser, 1989; Eisner & Meinwald, 1995; González et al., 1999; Torres-Vila, Rodríguez-Molina & Jennions, 2004; Torres-Vila & Jennions, 2005; Watanabe, 2016; Meslin et al., 2017; Cannon, 2020). The fact that most of these components are unavailable in the adult diet adds to their importance for female fitness and explains, in some extent, the ubiquity of polyandry in this group (Drummond III, 1984; Eberhard, 1985; Simmons, 2001; Sanchez, Hernandez-Baños & Cordero, 2011; Cannon, 2020). However, intriguingly, in Lepidoptera there are some monandrous species (Drummond III, 1984; Eberhard, 1985; Walters et al., 2012; Caballero-Mendieta & Cordero, 2013; Konagaya, Idogawa & Watanabe, 2020).

After mating, female butterflies of polyandrous species become sexually unreceptive for a period of time (Sugawara, 1979; Drummond III, 1984; Oberhauser, 1989; Oberhauser, 1992; Kaitala & Wiklund, 1995). During this refractory period, the sperm is transferred from the spermatophore to the spermatheca, its final storage place within the female. The resource-rich spermatophore is gradually digested within the CB (Drummond III, 1984; Oberhauser, 1992; Galicia, Sánchez & Cordero, 2008; Walters et al., 2012; Meslin et al., 2015; Plakke et al., 2015; Watanabe, 2016). At the proximate level, female sexual receptivity and mating frequency are controlled by multiple factors (Sugawara, 1979; Drummond III, 1984; Wedell, 2005). One important factor is the mechanical stimulation resulting from distension of the CB by the spermatophore (Labine, 1964; Sugawara, 1979; Oberhauser, 1992). Sugawara (1979) clearly demonstrated that reception of a spermatophore in the butterfly Pieris rapae (Pieridae) induces females to display mate rejection behaviour when courted and that stretch receptors on the surface of the CB are involved in this behavioural change. Sugawara (1979) showed that the frequency of afferent nervous impulses from the stretch receptors increase tenfold after reception of a spermatophore, however, females remain sexually receptive if the CB is filled with less than half the volume of an average spermatophore (recently and multiply mated males produce smaller spermatophores). In polyandrous species, female receptivity is gradually recovered as the amount of spermatophore remaining in the corpus bursa decreases (Oberhauser, 1989; Oberhauser, 1992) due to its digestion and absorption (Boggs & Gilbert, 1979; Lai-Fook, 1986; Galicia, Sánchez & Cordero, 2008; Meslin et al., 2015; Plakke et al., 2015).

Besides its effect on female receptivity, the presence of a spermatophore in the CB triggers the periodical contraction of the muscles surrounding the CB (Sugawara, 1979), resulting in tearing of the spermatophore envelope with the sclerotized structures located in the inner wall of the CB (called signa) and the mechanical digestion of the spermatophore contents (Sugawara, 1979; Rogers & Wells, 1984; Tschudi-Rein & Benz, 1990; Galicia, Sánchez & Cordero, 2008; Galicia, Sánchez & Cordero, 2014 ). The frequency of contractions of the CB muscles is directly correlated with the volume of the spermatophore (Sugawara, 1979). In species lacking signa, such as Calpodes ethlius (Hesperiidae), mechanical tearing and digestion of the spermatophore is also achieved via the “relatively violent” contractions of the muscles surrounding the CB (Lai-Fook, 1986: p. 556). The ubiquity of the mechanical digestion function of the CB is supported by transcriptomic studies showing highly expressed genes whose products are biased towards muscle organization and activity in the CB of P. rapae (Meslin et al., 2015) and the moth Ostrinia nubilalis (Crambidae) (Al-Wathiqui, Lewis & Dopman, 2014; Al-Wathiqui, Dopman & Lewis, 2016). More generally, the presence and layout of well-developed muscles surrounding the CB, and their common association with the signa, is consistent with a mechanical digestion function of the CB in Lepidoptera (Sugawara, 1979; Drummond III, 1984; Rogers & Wells, 1984; Kristensen, 2003; Lincango, Fernández & Baixeras, 2013). Spermatophore digestion is not only mechanical, but also biochemical, although this last process is less well understood (Al-Wathiqui, Dopman & Lewis, 2016). Transcriptomic and proteomic studies of the CB support the idea that the spermatophore is also enzymatically digested (Al-Wathiqui, Lewis & Dopman, 2014; Meslin et al., 2015; Al-Wathiqui, Dopman & Lewis, 2016) and pioneering studies of the CB of P. rapae show its proteolytic activity (Plakke et al., 2015) and have characterized some of the proteases involved (Plakke et al., 2019).

A third function of the CB is the absorption and transport of substances contained in the spermatophore. In his general review of lepidopteran genitalia, Kristensen (2003: p. 438) mentions the following: “The bursa is obviously capable of absorbing breakdown compounds from the spermatophore”. This function was demonstrated in C. ethlius (Lai-Fook, 1991) and is consistent with radiotracer studies in three nymphalid species (Boggs & Gilbert, 1979), transcriptomic studies in P. rapae identifying highly expressed genes associated to the transport function (Meslin et al., 2015), the presence of pores and the structure of epithelial cells in the monarch butterfly (Rogers & Wells, 1984), and observations of pores on the inner surface of the CB of several species of moths in the family Tortricidae (Lincango, Fernández & Baixeras, 2013; although these authors suggest pores could be involved in secretion of substances to the interior of the CB).

Summarizing, in recently mated female lepidopterans, the distension of the CB by the spermatophore turns-off sexual receptivity and triggers contractions of the muscles surrounding the CB that result in the piercing or tearing of the spermatophore envelope and the mechanical digestion of its contents (Sugawara, 1979; Lai-Fook, 1991). Female receptivity is recovered as the amount of spermatophore remaining in the CB decreases (Sugawara, 1979; Oberhauser, 1989; Oberhauser, 1992) due to digestion and absorption in the CB (Boggs & Gilbert, 1979; Lai-Fook, 1991; Galicia, Sánchez & Cordero, 2008; Meslin et al., 2015; Plakke et al., 2015; Watanabe, 2016). Thus, the CB plays a fundamental role in the control of sexual receptivity (although it is not the only factor; see Wedell, 2005) and mating frequency in female Lepidoptera and, therefore, it is an obvious place to look for genital adaptations to monandry in Lepidoptera (Cordero & Baixeras, 2015).

Here, we report observations regarding the possible mechanism responsible for female monandry in the butterfly Leptophobia aripa (Boisduval, 1836) (Pieridae, Pierinae). In the field, females of this species mate on average (SD) 1.19 (0.4) times, as judged from spermatophores counts in mated females (Caballero-Mendieta & Cordero, 2013). We studied the fate of the spermatophore within the CB with the aim of measuring its rate of digestion and, surprisingly, found that the spermatophore does not show any sign of being digested. To shed light on why the spermatophore is not digested, we studied the musculature of the CB, as well as the fine structure of its inner surface. We found that L. aripa lacks the necessary “apparatus” for the mechanical digestion of the spermatophore.

Materials and Methods

Butterflies studied and laboratory rearing

L. aripa is the most abundant butterfly in Mexico City, flying all year (Díaz Batres & Llorente Bousquets, 2011). Their caterpillars feed on a variety of plant species and are considered a pest of cabbage, broccoli and cauliflower crops in México and Central America (CATIE/MIP, 1990). The butterflies used in our experiments and in most observations were the offspring of females collected in the Ciudad Universitaria campus of the Universidad Nacional Autónoma de México (CU-UNAM), located in southern Mexico City. Individual females were fed ad libitum every morning a 10% sugar solution and allowed to lay eggs in plastic containers with fresh leaves of Tropaeolum majus (Tropaeolaceae), the main food plant in our study location. To stimulate oviposition, the containers, covered with mesh cloth, were located under (about 20 cm) an incandescent white light bulb for 90 min, although females frequently lay eggs even in the absence of these bulbs. The larvae were reared individually in small plastic containers (10 cm diameter, four cm height) with T. majus fresh leaves. Upon emergence, adults were individually marked on the wings with a permanent marker (Sharpie™) and kept individually in the same plastic containers in which they were reared.

Experiment on the fate of the spermatophore within the corpus bursae

This experiment was originally designed to determine the pattern of digestion of the spermatophore within the CB. Virgin females were mated with virgin males and euthanized by freezing at different times after the end of copulation (0, 8, 16, 24, 48, 72 and 96 h). Matings were obtained by placing males and females in cylindrical cages made of mesh cloth and metal wire (∼60 cm height and ∼25 cm diameter) in the gardens of the Instituto de Ecología, located in CU-UNAM, between 10AM and 15PM (Mexico City time). With exception of the females frozen immediately after finishing copulating (0h), all experimental females were allowed to lay eggs daily as explained above. All females laid eggs and in most cases these were numerous, although they were not counted.

The frozen females were thawed at ambient temperature and their abdomens separated from the body, opened and cleaned with forceps. Then, the CB and the spermatophores were carefully dissected out, thoroughly examined and photographed under a stereomicroscope (Olympus™ BX 51). A total of 48 females were studied: N0h = 6 females, N8h = 7, N16h = 7, N24h = 8, N48h = 8, N72h = 6, and N96h = 6. These times were chosen because females under laboratory conditions lay most of their eggs within four or five days after mating, and few live more than a week, despite being fed daily (D. Xochipiltecatl, per. obs.). In each spermatophore photograph, we measured the area covered by the spermatophore with the ImageJ open access software (National Institutes of Health USA, http://rsb.info.nih.gov.ij/). We used this area as a proxy of spermatophore size. We compared the effect of time after mating on the area of the spermatophore with a Kruskal–Wallis ANOVA.

Preparation of samples for microscopic observation

Observations and photographs of the CB, the ductus bursae (the duct connecting the CB with the copulatory pore known as the ostium; Fig. 1) and the spermatophore, of dry and fixed specimens (see below), were made with stereomicroscopes (Olympus™ BX 51 and Leica™ MZ8) and a scanning electron microscope (SEM; Hitachi™ S4800).

Figure 1 Female genitalia of the butterfly Leptophobiaaripa.

(A) Genitalia of a virgin female: note the empty corpus bursae (cb) and the signum (si) near the junction (cervix) with the ductus bursae (db). (B) Corpus bursae and ductus bursae of a mated female: note the spermatophore almost filling the corpus bursae and the collum of the spermatophore filling the ductus bursae. Scale bars A = 500 µm; B = 1,000 µm.

For observation of the muscles associated with the CB and ductus bursae (DB hereafter) we used two methods. First, three laboratory-reared virgin females were placed in a freezer at −20 °C for about 3 min and then gently injected with Karnovsky’s fixative (paraformaldehyde 2%/glutaraldehyde 2.5%) in the body cavity through the thorax and the abdomen. Then the abdomen was separated from the rest of the body and submerged in the same fixative until dissection. For SEM observation, the abdomens were transferred to centrifuge tubes with phosphate buffer 0.1M and rinsed during several minutes in a shaker (MRS-Mini Rocket Shaker, Biosan™). Then, the abdomens were carefully removed and cleaned with forceps, and the CB and DB were dissected out, stained with 2% osmium tetroxide for 20 min followed by thoroughly washing with water, placed in microporous specimen capsules (30 µm pore size, Ted Pella Inc., product number 4619) and dehydrated in increasing grade ethanol. The CB and DB were then dried to critical point in an Autosamdry 814™ (Tousimis), positioned on SEM stubs using carbon tape and silver conducting paint and sputtered with Au-Pd. For comparison purposes, similar procedures were applied to two females of the common African leafworm, Spodoptera littoralis (Boisduval, 1833). This noctuid species allows easy recognition of the musculature associated to the CB.

We also observed the muscles associated to the CB and DB in three field collected mated females (captured while laying eggs) that were brought to the laboratory and allowed to continue laying eggs (one female two days and two females three days), before being euthanized by freezing at −70 °C and then their abdomens were separated from the body and preserved in 100% ethylic alcohol. The abdomens were carefully opened and the CB and DB dissected out and carefully cleaned, with micro-scissors, fine forceps and fine brushes, in glass embryo dishes under the stereomicroscope. The spermatophores contained in these females were also used to confirm that they are not digested in the CB (see Results).

For observation of the inner surface of the CB in the SEM, we used three laboratory-reared virgin females, two of them preserved dry and one fixed (with paraformaldehyde 2%/glutaraldehyde 2.5%) as explained above. The abdomen of the fixed specimen was rinsed in phosphate buffer 0.1M, as explained above. The abdomens were separated from the rest of the body and digested in 10% KOH at 90 °C for about 90 min. Then, the abdomens were stained for about 30 s in chlorazol black (0.1% in ethanol 70°) and the CB was removed, carefully cleaned and cut longitudinally. Subsequent digestions with KOH were performed when needed. Fragments were processed in a similar way to the treatment of complete CB and DB.

Results

The spermatophore is not digested in the corpus bursae

Forty-eight virgin females were mated with virgin males and frozen at different times after the end of copulation. These females had their spermatophores carefully dissected and thoroughly examined under the dissection microscope. Despite the fact that most females laid eggs before being dissected (the exception being the females frozen immediately after mating), the spermatophores contained in the CB of all females remained physically intact independently of the time elapsed after the end of copulation (Fig. 2). There was no sign of rupture on the external surface of the spermatophore or of spermatophore deflation (Fig. 2). In agreement with these observations, the area covered by the spermatophore, our proxy of spermatophore size, did not vary with time lapsed after copulation (Kruskal–Wallis test: H = 4.26, P = 0.64, df = 6; Fig. 3).

Figure 2 The spermatophores are not digested within the corpus bursae of female Leptophobia aripa butterflies.

Typical examples of spermatophores showing that they remain intact in the CB independently of the time elapsed after the end of copulation (number of hours written besides each photograph) and of the fact that females laid eggs. Scale bar = 1 mm.

Figure 3 Area covered by the spermatophore (a proxy of its size) as a function of time elapsed after the end of copulation.

For each time, the Median (black line), 25% and 75% quartiles (grey box) and minimum and maximum values (whiskers) are shown (an outlier was detected at 24 h). Time after copulation had no effect on spermatophore area (Kruskal–Wallis test: H = 4.26, P = 0.64, df = 6).

The bulbous spermatophore occupies most of the CB (Fig. 1B), is bilobed (Fig. 4A) and has a tubular prolongation, called the collum, which extends along the DB (Figs. 1B, 2 and 4A), blocking it almost completely (Fig. 1B). Only the area of the spermatophore that is in contact with the signum looked somewhat deformed but not broken (Fig. 4A).

Figure 4 SEM images of the spermatophore, bursa copulatrix and its associated muscles, of the butterfly Leptophobia aripa and the spruce cotton leafworm Spodoptera littoralis.

(A) An intact spermatophore obtained from a female of L. aripa collected while laying eggs, taken to the laboratory and allowed to continue laying eggs for two more days; notice the tubular collum. (B) CB and DB of L. aripa showing the muscular lining of the DB; the CB appears corrugated, with no muscle vestiture. (C) CB and DB of the African cotton leafworm showing a complex muscular lining; detail of the muscular area in the white rectangle is shown in F. (D) Inner surface of the CB of L. aripa showing a complete absence of pores. (E) Close-up of the “junction” area of the CB and the DB (the cervix) in L. aripa showing muscle fibers covering the DB, absent on the CB. (F) Detail of the muscle lining of the CB of S. littoralis marked by the white rectangle in C. Abbreviation: co, collum. Scale bars A and B = 500 µm, C = 1.000 µm, D = 5 µm, E = 150 µm, F = 100 µm.

These observations were corroborated with detailed SEM observations of the spermatophores obtained from the three females collected in the field while laying eggs. As observed in the previous experiment, the external envelopes of these spermatophores were also intact although somewhat compressed near the signum (Fig. 4A). No deflation, perforations or tearing were observed in any of them (Fig. 4A). Thus, our observations indicate that the spermatophore is not digested at least up to four days after mating, which is enough time for females to lay most of their eggs under laboratory conditions (D. Xochipiltecatl, pers. obs. of oviposition of several laboratory-reared females in 2016 and 2017).

The corpus bursae lacks the necessary “apparatus” for the mechanical digestion of the spermatophore

The observation of the CB of Lepidoptera under the optical microscope allows detection of sclerotizations (signa) and folds. The observation under SEM of the external surface of the CB of many species (such as S. littoralis) reveals the presence of a muscular lining over the integument. Muscles appear as bundles of parallel fibers (Figs. 4C and 4F) easily distinguishable from folds and other structural components. The integument of the CB of L. aripa through the optical microscope (Fig. 1A) appears rather uniform, smoothly rough, and somewhat transversally corrugated. Except for the presence of the signum near the junction area of the DB and CB (known as the cervix bursae), no sclerotization is detectable. Observations through SEM of the CB of virgin and mated females of L. aripa (Figs. 4B and 4E) fixed either with formaldehyde/glutaraldehyde (N = 3) or with absolute ethylic alcohol (N = 3) did not show muscles enveloping the CB (Fig. 4B), thus indicating that mechanical digestion of the spermatophore is not feasible in this species. The integument is directly exposed on the CB and follows the corrugation observable through the optical microscope. The DB, on the contrary, is covered with muscle fibers extending from a ventromedial line, which gives its superficially striated appearance (Figs. 4B and 4E). Some of these fibers are inserted in the cervix, at the level of the signum. They could help this structure to exert pressure and deform the spermatophore without breaking it (Fig. 4A).

Detailed observations in the SEM of the inner surface of the CB showed a complete absence of pores (Fig. 4D).

Discussion

In this paper, we show that in the butterfly L. aripa the spermatophore remains intact within the CB at least up to four days after mating (Fig. 2), the period during which females lay most of their eggs in captivity (D. Xochipiltecatl, pers. obs. of oviposition of several laboratory-reared females in 2016 and 2017). In agreement with these observations, the area covered by the spermatophore (a proxy of its size) did not decrease with the time lapsed after the end of copulation (Fig. 3). These observations indicate that in this butterfly females do not digest the spermatophores, in contrast with most lepidopterans studied (Drummond III, 1984; Oberhauser, 1992; Galicia, Sánchez & Cordero, 2008; Walters et al., 2012; Meslin et al., 2015; Plakke et al., 2015; Watanabe, 2016). We also show that there are no muscles enveloping the CB. Since in other lepidopterans the spermatophore is mechanically digested due to the contractions of the muscular sheath of the CB (Sugawara, 1979; Rogers & Wells, 1984; Lai-Fook, 1986; Al-Wathiqui, Lewis & Dopman, 2014; Meslin et al., 2015), we propose that the absence of a muscular sheath prevents the CB from digesting mechanically the spermatophore. In other words, females of this species lack the “apparatus” required for the mechanical digestion of spermatophores. Furthermore, judging from the intact condition of the spermatophores several days after mating in females that laid eggs during several days, enzymatic digestion also seems to be absent. The observed absence of pores on the inner surface of the CB is also consistent with this idea because pores could be involved both in the absorption of products from digestion of spermatophores (Lai-Fook, 1986), and in the secretion of molecules used for the chemical digestion of spermatophores within the CB, as suggested for Tortricidae (Lincango, Fernández & Baixeras, 2013).

In the introduction, we reviewed evidence indicating that in many Lepidoptera female sexual receptivity is at least partially controlled by the mechanical stimulation (distension) of the CB by the spermatophore. The degree of distension of the CB is inversely related to the sexual receptivity of females, and the receptivity is recovered as the ejaculate is digested and the CB deflates (Labine, 1964; Sugawara, 1979; Oberhauser, 1992). L. aripa females tend to be monandrous (Caballero-Mendieta & Cordero, 2013) and we propose that female monandry in this species is a result of its incapability to mechanically digest the spermatophore, which results in a constant degree of CB distension after mating and, thus, in the maintenance of the sexually unreceptive state of females. Thus, we propose that the absence of muscles enveloping the CB explains monandry in L. aripa. A possible explanation for the rare cases of twice-mated females in this species (Caballero-Mendieta & Cordero, 2013) is that their first mate was recently mated and/or small, since both male conditions are known to result in the transfer of smaller spermatophores (Caballero-Mendieta & Cordero, 2013). This hypothesis can be tested by comparing remating rates of females mated to virgin/average sized males with those of females mated with recently mated/small males.

As mentioned in the Introduction (references therein), many male lepidopterans produce spermatophores rich in nutrients and other chemical compounds that enhance female fitness. Since these “nuptial gifts” are costly to produce (Shapiro, 1982; Cordero, 2000; Ferkau & Fischer, 2006; Caballero-Mendieta & Cordero, 2013), we predict that in L. aripa selection acting on males favours a reduction in the content of nutrients in the spermatophore in comparison with species in which females digest the spermatophore and obtain fitness benefits (see references in the Introduction). This prediction can be tested by comparing the relative content of nutritious substances in the spermatophore of L. aripa and in species in which females digest spermatophores.

According to Drummond, “In some short-lived temperate zone butterflies, the spermatophore is known to persist intact in laboratory-held females for longer than the life expectancy of a female in the wild” (Drummond III, 1984: p. 303). We predict that these species are monandrous and possibly have a CB devoid of a muscular sheath. Also, it will be interesting to study if the CB of known monandrous butterflies and moths (Drummond III, 1984; Sanchez, Hernandez-Baños & Cordero, 2011; Konagaya, Idogawa & Watanabe, 2020) lacks a muscular sheath. In this case, it will be particularly interesting the comparison of the monandrous pupal-mating Heliconius with the polyandrous species of the same genus (Walters et al., 2012; Jiggins, 2017).

There are two general hypotheses to explain the evolutionary origin and maintenance of monandry in insects (Arnqvist & Nilsson, 2000; Arnqvist & Andrés, 2006; Hosken et al., 2009): either monandry is selected for in females when they maximize their fitness with just one mating, or sperm competition favours male adaptations that impose monandry on females that, otherwise, could obtain benefits from multiple mating. We suggest that the absence of a key adaptation required for the mechanical digestion of spermatophores sheds light on the selective pressures that favoured monandry in L. aripa. The muscular sheath of the CB is generally associated to the presence of signa (Sugawara, 1979; Drummond III, 1984; Rogers & Wells, 1984; Lai-Fook, 1986; Kristensen, 2003; Lincango, Fernández & Baixeras, 2013; Al-Wathiqui, Lewis & Dopman, 2014; Meslin et al., 2015) and signa appear to be a general feature of Lepidoptera (Sanchez, Hernandez-Baños & Cordero, 2011). Thus, although a proper phylogenetic study is required, we hypothesize that the muscular sheath was lost in L. aripa and that this loss is a female adaptation to monandry.

Why monandry could be adaptive for females in this species remains to be studied. One interesting hypothesis is that increases in the availability of nutrients in food plants have reduced the importance of spermatophore-derived nutrients for female reproduction and favoured monandrous mating in females, possibly because in this way females reduce copulation time costs and predation risk during courtship and copulation. A recent study proposed this idea and presented evidence that anthropogenic nutritional enrichment of food plants has an effect on female mating frequency (Espeset et al., 2019). A comparison of an “agricultural population” (AP) of the butterfly Pieris rapae, where fertilizers, irrigation and low levels of pesticides resulted in increased availability of nitrogen in food plants (canola), with a non-agricultural population (NAP) showed that, as predicted, most females of the AP mated once whereas more than half of the females of the NAP mated two or three times (Espeset et al., 2019). In the case of L. aripa in our study site, the females lay eggs mostly in a non-cultivated plant (T. majus) that grows forming large patches in disturbed places like the side of roads, but also grows within the gardens of the University, where it at least receives irrigation. On the other hand, in other parts of the city, this butterfly uses as host plants cultivated vegetables that can be fertilized and are irrigated (such as cabbage, broccoli and cauliflower; CATIE/MIP, 1990). A second hypothesis is that monandry evolved in response to male adaptation to sperm competition. For example, if males evolved spermatophores that are difficult to digest to delay female remating, a point could be reached in which spermatophore digestion becomes excessively expensive due to physiological or ecological reasons, and favours females that avoid these costs by abandoning polyandry.

Supplemental Information

Supplemental Information 1 Area covered by the spermatophores in photographs taken under a stereomicroscope

The areas were measured with the open access software ImageJ (National Institutes of Health USA, http://rsb.info.nih.gov.ij/). Time is the number of hours after the end of copulation when the female abdomen was fixed. Spermatophore area is given in squared millimetres.

Click here for additional data file.

The research reported in this paper is part of the doctoral thesis of David Xochipiltecatl García (DXG) in the Posgrado en Ciencias Biológicas, Universidad Nacional Autónoma de México (UNAM). We thank Dr. Juan Núñez Farfán and Dr. Jorge Contreras for valuable criticism and suggestions on the project, and Raúl Iván Martínez for technical support. We thank the students of Carlos Cordero’s lab for discussion and support. We thank Dr. Boyan Zlatkov, Dr. Andrew Stoehr and an anonymous reviewer for insightful commentaries that helped us to improve our manuscript.

Additional Information and Declarations

Competing Interests

Author Contributions

Data Availability

The authors declare there are no competing interests.

David Xochipiltecatl conceived and designed the experiments, performed the experiments, analyzed the data, prepared figures and/or tables, authored or reviewed drafts of the paper, and approved the final draft.

Joaquín Baixeras performed the experiments, analyzed the data, prepared figures and/or tables, authored or reviewed drafts of the paper, and approved the final draft.

Carlos R. Cordero conceived and designed the experiments, analyzed the data, authored or reviewed drafts of the paper, and approved the final draft.

The following information was supplied regarding data availability:

The raw spermatophore area measurements are available in the Supplementary File.

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
