# Peer review of "Atypical functioning of female genitalia explains monandry in a butterfly"

_PeerJ, doi:10.7717/peerj.12499_

## Round 0.1 · original submission · Major Revisions

Dear Dr. Xochipiltecatl and colleagues:

Thanks for submitting your manuscript to PeerJ. I have now received three independent reviews of your work, and as you will see, the reviewers raised some concerns about the paper. Despite this, this the reviewers are optimistic about your work and the potential impact it will have on research studying monandry and sexual selection. Thus, I encourage you to revise your manuscript, accordingly, taking into account all of the concerns raised by the reviewers.

While the concerns of the reviewers are relatively minor, this is a major revision to ensure that the original reviewers have a chance to evaluate your responses to their concerns.

Importantly, please make clear your speculations in the paper. Please also provide some future directions for the field. Please make sure all relevant information and references are provided (especially figures).

There are many minor suggestions to improve the manuscript (typos, nuances, etc.). Reviewer 1 has kindly provided edits on your manuscript.

I agree with many of the concerns of the reviewers, and thus feel that their suggestions should be adequately addressed before moving forward.

I look forward to seeing your revision, and thanks again for submitting your work to PeerJ.

Good luck with your revision,

-joe

·

Basic reporting

The study is focused on an unusual condition (lacking muscularis on corpus bursae) of the female genitalia of a butterfly, related to monandry. The text is well written, concise and clear. All relevant literature sources are cited. The figures demonstrate well the results and are of very good quality.
Following the tips, I can confirm:
Clear and unambiguous, professional English used throughout. YES
Literature references, sufficient field background/context provided. YES
Professional article structure, figures, tables. YES
Self-contained with relevant results to hypotheses. YES

Experimental design

The research is original and is focused on an important but rarely studied field of the sexual selection. The monandry is a rare phenomenon among Lepidoptera and the present study reveal the reasons for its existence in a model species. All ethical standards applicable to the studied group of organisms are followed. The methods used in the study are appropriate and are described in detail in the Material and methods section, ensuring reproducibility of the results.

I confirm thefollowing:
Original primary research within Aims and Scope of the journal.
Research question well defined, relevant & meaningful. It is stated how research fills an identified knowledge gap.YES
Rigorous investigation performed to a high technical & ethical standard. YES
Methods described with sufficient detail & information to replicate. YES

Validity of the findings

The results are clearly explained. They are commented in the Discussion without any excessive speculations.

All underlying data have been provided; they are robust and controlled. YES
Conclusions are well stated, linked to original research question & limited to supporting results. YES
Speculation is welcome, but should be identified as such. Speculations without supporting results are not formulated in the paper.

Additional comments

Though undesirable, I would like to say that the study is of particular interest for me. The problems of the sexual selection and related phenomena are of fundamental importance for understanding the evolution in general. I believe that the paper will be of interest for most biologists, not only entomologists. I made some minor suggestions in the manuscript (mostly editorial), as well as a comment, using the editing mode of Word. I recommend publication of the manuscript after minor corrections, without any extensive changes.

·

Basic reporting

Because this manuscript is very short and does not really involve "experiments" or "data analysis", I have simply put all of my detailed comments here in the "Basic reporting" section. I hope that is OK.

This short manuscript describes some interesting (and possibly unique) anatomical features of a (mostly) monandrous butterfly that might help explain the mating behavior of the species.

I very much enjoyed this manuscript. The paper is very clearly written and concise, but at the same time appropriately detailed. The opening arguments (lines 34-124) very clearly make a compelling case for why the findings to come later are of interest. Although the paper does not make the point explicitly, it is a very good example of what can come of the appropriate overlap of basic natural history, anatomy and physiology, and evolutionary biology. In addition to being interesting and well-written, the figures are very nice.

I have really no substantive criticisms of the paper and only a few suggestions.

1) First, if photos, with scale-bars, of all of the dissected spermatophores exist would it be possible to provide some basic summary statistics of the sizes of the spermatophores? If the photographs are not sufficiently standardized then maybe this wouldn’t be appropriate. However, in addition to strengthening the case for their hypothesis some summary statistics could be useful for future work, whether in this species or others, that might wish to conduct similar analyses.

2) Second, could the authors propose some future experimental manipulations that might further support their hypothesis? I have on mind a couple of ideas although I admit I have not thought these through in detail, nor do I know if they are actually possible. First, do methods exist that would allow one to inject the bursa of females with something that would gradually “digest”? If so, you might predict that this simulation of a digesting spermatophore would result in the initial loss of, and eventual resumption of, female receptivity. This could show that monandry is indeed a function of activities inside the bursa and related to the physical size of the spermatophore and not some other cause. Second, could females be mated to virgin males and recently mated males to show that smaller spermatophores do indeed fail to induce monandry? I should add that I am not suggesting that this paper needs to do these experiments; I am just wondering if the authors have thoughts, beyond the phylogenetic analyses they already have proposed, for further work that could investigate the ideas in the future.



3) I have also included some very minor suggestions about wording changes that would make some sentences more consistent with conventional English. In particular, there are several places where the article “the” should probably be inserted. Those, and some other instances, are reported below by line number.


L97: “More generally…” would be more appropriate than “More general…”

L105: “mentions the following: ‘The bursa…” is more appropriate, since a colon should be preceded by a complete clause.

L34-124: This opening argument is very well written. It is clear and sets up the purpose of the study nicely.

L170: Add “the”, as in “known as the ostium”

L173: I think “associated with” instead of “associated to” is a bit more conventional in English.

L175: “injected with Karnovsky’s fixative”

L187: “allowed to continue laying eggs”

L212: “called the collum”

L261: “is a result of”

L283: “signa appear” (instead of appears, since signa is the plural)

L300: “where it at least receives irrigation”

Experimental design

Please see my comments in the "basic reporting" section. The research approach is appropriate, as I explained, with only some minor suggestions.

Validity of the findings

Please see my comments in the "basic reporting" section. The authors do a very good job arguing for the validity and novelty of their findings.

Reviewer 3 ·

Basic reporting

“Atypical functioning of female genitalia explains monandry in a butterfly” by Xochipiltecatl et al documents the physiological abnormalities on the corpus bursae of Leptophobia aripa. This study demonstrates that the CB does not puncture and digest the spermatophore as is common in polyandrous species, and that the CB itself appears devoid of muscles.

Previous studies have shown that the presence/absence of the signa correlates with the propensity for a female to remate, but this study is the first to my knowledge that observes a lack of muscle on the CB which could potentially lead to the inability of females to digest the spermatophore. Such a finding is exciting and has the potential to shed light on the sexual antagonistic coevolution that is common across the Pieridae clade.

The article was presented in a straightforward manner with a clear narrative that was simple to understand and follow. There were, however, a few places in the manuscript in which sentence structure and grammar could be revisited for improved clarity:
• Line 174 “introduced in a freezer” might be better phrased as “placed in a freezer” or alternatively “frozen at”
• Line 181 “during 20 minutes” could be “for 20 minutes”
• Line 187-188 “continuing laying eggs” instead might be “to continue laying eggs”
• Line 205-207 this is a very long sentence, I recommend breaking it apart for increased clarity
• Line 300 “where at least receives irrigation”. I’m unclear what this statement is supposed to mean, please consider rephrasing

As far as literature references and background go, this manuscript provides a robust reference base for the system and previous knowledge (though would benefit from citing literature on the European Corn Borer conducted by Al-Wathiqui et al). However, the discussion does make note of enzymatic digestion of the spermatophore in combination with mechanical digestion (line 250) yet the introduction does not provide background on alternative digestion of the spermatophore except for mechanical. I would suggest including some background on the various mechanisms used to digest the spermatophore in order to be able to more completely discuss the lack of digestion observed.

Experimental design

My one concern about this manuscript is the general lack of data and quantification for the phenotypes observed. For example, visually it appears that the spermatophores are not being digested, but are there any quantifiable measures for this? For example the surface area or mass of the spermatophores across numerous individuals in order to show that no digestion is occurring?

Second, while it seems like there are no muscles on the CB, are there any ways to verify this? I would personally recommend staining for muscle in order to show lack of staining on the CB in contrast to likely strong staining on the ductus bursa.

Validity of the findings

I will echo my comments above that there are no quantifiable data provided, which would greatly increase the strength of the manuscript.

As far as the conclusions for the manuscript, I appreciated the proposed hypotheses that the lack of muscle is a female adaptation to monandry. I find it strange that there was no comparison to or contrast to the systems in which signa have been lost, and further still that in a system where females no longer digest the spermatophore, that males would continue to produce such a costly gift. Perhaps this could be addressed in the discussion?

---

## Round 0.2 · accepted · Accept

Dear Dr. Xochipiltecatl and colleagues:

Thanks for revising your manuscript based on the concerns raised by the reviewers. I now believe that your manuscript is suitable for publication. Congratulations! I look forward to seeing this work in print, and I anticipate it being an important resource for groups studying monandry and sexual selection. Thanks again for choosing PeerJ to publish such important work.

Best,

-joe

·

Basic reporting

Since I reviwed the manuscript at the first review round, I am not going to repeat my review but I write on some tips that I supopse should be mentioned, generally following the instructions:

Clear and unambiguous, professional English used throughout. YES
Literature references, sufficient field background/context provided. YES
Professional article structure, figures, tables. YES; the authors added a figure of a bursa with developed muscularis and now even non-specialists can realise what is it;
Self-contained with relevant results to hypotheses. YES

Experimental design

Study designed and performed excellently.

Original primary research within Aims and Scope of the journal. YES
Research question well defined, relevant & meaningful. It is stated how research fills an identified knowledge gap.YES
Rigorous investigation performed to a high technical & ethical standard. YES
Methods described with sufficient detail & information to replicate. YES

Validity of the findings

The results are clearly explained. They are commented in the Discussion without any excessive speculations.
All underlying data have been provided; they are robust and controlled. YES
Conclusions are well stated, linked to original research question & limited to supporting results. YES

Additional comments

The authors addressed all comments of the reviewers, and extended some paragraphs accordingly. I would be happy to see this work published.

·

Basic reporting

As I said in my first review, my most substantive comment was a request for some quantitative data and the authors have now provided that, so I find this version of the manuscript satisfactory. The other changes made in response to my comments and those of the other reviewers also improved the manuscript, in my opinion.

Experimental design

See comments in "Basic reporting" section.

Validity of the findings

See comments in "Basic reporting" section.

Additional comments

See comments in "Basic reporting" section.

Reviewer 3 ·

Basic reporting

No comment

Experimental design

No comment

Validity of the findings

No comment

Additional comments

The manuscript, after undergoing revisions, has been updated to address all concerns I had previously. I have no further comments or suggestions for improvement of the manuscript!